# Phytochemicals in Inhibition of Prostate Cancer: Evidence from Molecular Mechanisms Studies

**DOI:** 10.3390/biom12091306

**Published:** 2022-09-16

**Authors:** Qiongyu Hao, Yanyuan Wu, Jaydutt V. Vadgama, Piwen Wang

**Affiliations:** 1Division of Cancer Research and Training, Department of Internal Medicine, Charles R. Drew University of Medicine and Science, Los Angeles, CA 90059, USA; 2Jonsson Comprehensive Cancer Center, David Geffen School of Medicine, University of California at Los Angeles, Los Angeles, CA 90095, USA

**Keywords:** natural products, chemotherapy, prostate cancer, mechanisms studies

## Abstract

Prostate cancer is one of the leading causes of death for men worldwide. The development of resistance, toxicity, and side effects of conventional therapies have made prostate cancer treatment become more intensive and aggressive. Many phytochemicals isolated from plants have shown to be tumor cytotoxic. In vitro laboratory studies have revealed that natural compounds can affect cancer cell proliferation by modulating many crucial cellular signaling pathways frequently dysregulated in prostate cancer. A multitude of natural compounds have been found to induce cell cycle arrest, promote apoptosis, inhibit cancer cell growth, and suppress angiogenesis. In addition, combinatorial use of natural compounds with hormone and/or chemotherapeutic drugs seems to be a promising strategy to enhance the therapeutic effect in a less toxic manner, as suggested by pre-clinical studies. In this context, we systematically reviewed the currently available literature of naturally occurring compounds isolated from vegetables, fruits, teas, and herbs, with their relevant mechanisms of action in prostate cancer. As there is increasing data on how phytochemicals interfere with diverse molecular pathways in prostate cancer, this review discusses and emphasizes the implicated molecular pathways of cell proliferation, cell cycle control, apoptosis, and autophagy as important processes that control tumor angiogenesis, invasion, and metastasis. In conclusion, the elucidation of the natural compounds’ chemical structure-based anti-cancer mechanisms will facilitate drug development and the optimization of drug combinations. Phytochemicals, as anti-cancer agents in the treatment of prostate cancer, can have significant health benefits for humans.

## 1. Introduction

Prostate cancer (PCa) is the second most commonly diagnosed cancer and the fifth leading cause of cancer death among men worldwide [1]. The American Joint Committee on Cancer (AJCC) developed the TNM (tumor, node, metastasis) system to describe the five stages of prostate cancer: Stage 0, and Stages I to IV. Hormone-naïve advanced prostate cancer is very sensitive to hormones; thus, androgen deprivation therapy (ADT) has been used as the conventional therapy for advanced PCa [2]. Hormone naïve advanced prostate cancer will eventually progress to castration-resistant prostate cancer (CRPC) within 2–3 years after the initial ADT treatment [3]. With an improved understanding that the growth of most CRPC tumors is still dependent on androgen and androgen receptors (AR), several second-generation antiandrogens, such as abiraterone acetate (preventing androgen biosynthesis) and enzalutamide (preventing AR translocation to the nucleus), were developed. However, resistance to these antiandrogens has also appeared clinically [4]. After the failure of antiandrogens, patients will typically be treated with chemotherapy, such as docetaxel/prednisone treatment. Although some patients are initially responsive to chemotherapy, resistance usually occurs in 1–2 years [5]. Traditional herb medicine has used plants to treat various diseases for many years. A number of natural products have been isolated from plants to test their tumor cytotoxic efficacy in the last few decades. The phytochemicals’ natural and raw forms are important for the effectiveness of the agents. Laboratory studies have revealed that natural compounds can affect cellular proliferation by modulating many crucial cellular signaling pathways. Given that the magnitude of decreasing PCa mortality, in company with increasing PCa incidence due to the adoption of screening-based early detection, has been attenuated globally in recent years [1], and both hormone therapy and chemotherapy have potential resistance and severe side effects, there is an intensive need to develop less toxic, practical approaches to improve these conventional treatments. Increasing evidence from laboratory studies has suggested that natural compounds are potential to enhance the therapeutic effect of hormone or chemotherapy in prostate cancer in a non-/less-toxic manner through selectively targeting diverse dysfunctional molecular pathways in cancer cells.

As there is increasing data on how phytochemicals interfere with diverse molecular pathways in PCa, this review discusses and emphasizes the implicated molecular pathways of cell proliferation, cell cycle control, apoptosis, and autophagy as important processes that control tumor angiogenesis, invasion, and metastasis. We reviewed the latest progress on phytochemicals in prostate cancer studies, being particularly interested in information on their mechanism of action. The phytochemicals and their major molecular targets are summarized in Table 1, illustrating the effectiveness of phytochemicals on the major signaling pathways involved in prostate cancer. Some detailed information for each of the phytochemicals is described below.

## 2. Polyphenols

### 2.1. Polyphenol Flavonol

#### 2.1.1. Quercetin

Quercetin is a bioactive flavonol pigment present in onions and apples. Quercetin augmented TRAIL-mediated PCa cell apoptosis and interfered with many oncogenes and tumor suppressor genes, showing cancer-protecting effects [141,142]. Quercetin provoked PCa cell apoptosis by inhibition of FA synthase and downregulation of Hsp90 [7]. Further, quercetin also could inhibit PCa stem cells by PI3K/Akt and MAPK/ERK signaling [6]. Moreover, quercetin exacerbated TRAIL-induced cytotoxicity by a reduction in survivin and Akt phosphorylation and caspase activation [143]. PCa cell xenograft mouse models demonstrated quercetin interacted with the VEGF-R2-regulated autophagic pathway, depicting its anti-angiogenetic effects [144].

#### 2.1.2. Apigenin

Apigenin is a flavone extracted from *Anthemis* spp. Apigenin suppressed cell proliferation by inhibiting the PI3k/Akt pathway and induced apoptosis by inactivating the IGF-IGF-IR signaling in PC-3 cells [8]. Apigenin also inhibited class I HDACs as a mediator of epigenetic events in PC-3 and 22Rv1 cells [9]. Apigenin-activated p53 pathways by ROS generation induced apoptosis in 22Rv1 cells [10]. Apigenin has shown an anti-angiogenetic potential by reduced vascular endothelial growth factor (VEGF) production in PC-3, LNCaP, and C4-2B cells, attenuating cancer progression and metastasis [11]. Additionally, apigenin inhibited PC-3 and 22Rv1 xenografts in athymic nude mice [145]. Apigenin also suppressed cancer progression in TRAMP mice [146].

#### 2.1.3. Baicalin

Baicalin is a flavone extracted from several species in the genus *Scutellaria*, such as *Scutellaria baicalensis* and *Scutellaria lateriflora*. Baicalein induced apoptosis and inhibited metastasis through inhibition of the caveolin-1/AKT/mTOR pathway in DU 145 and PC-3 cells [12]. Baicalein effectively suppressed the growth of AR-positive PCa cells, by inhibiting the AR N/C dimerization and AR-coactivators interaction in LNCaP cells and CWR22Rv1 cells [13]. Baicalein inhibited PC-3 cell proliferative activity by downregulating Ezrin [14].

#### 2.1.4. Cyanidin-3-Glucoside

Cyanidin-3-glucoside (C3G) is a major flavonoid anthocyanin in plant-based foods, such as leafy vegetables, berries, red cabbages, teas, and colored grains. It is a well-known natural anthocyanin and possesses anti-oxidant and anti-inflammatory properties. Jongsomchai et al. have demonstrated that C3G delays progressive cancer cell behaviors by inhibiting EMT through mediating Snail/E-cadherin expression [15].

#### 2.1.5. Daidzein

Daidzein is an isoflavone extracted from soybeans. Daidzein elicited cell cycle arrest by modulations of the CDK-related genes and a reduction in EGF and IGF in LNCaP, PC-3, and DU 145 cells [16]. Daidzein also caused epigenetic modifications of tumor suppressor genes, such as CpG island demethylation, thus showing a therapeutic role [17]. Novel daidzein molecules exhibited anti-prostate cancer activity through nuclear receptor ERβ modulation [18].

#### 2.1.6. Delphinidin

Delphinidin is an anthocyanidin extracted from *Viola* spp. Delphinidin inhibited NF-κB signaling and the activation of the subsequent caspase, leading to cell growth inhibition and apoptosis induction in a dose-dependent manner in LNCaP, C4-2, 22Rv1, and PC-3 cells [19]. Delphinidin inhibited cell growth through modulation of β-catenin signaling [20]. Delphinidin sensitized prostate cancer cells to TRAIL-induced apoptosis by inducing DR5 [21]. Delphinidin also induced p53 acetylation-mediated apoptosis by suppressing HDAC activity in LNCaP cells [22]. In addition, delphinidin administration markedly suppressed PC-3 xenografts growth in athymic nude mice [147].

#### 2.1.7. Epigallocatechin-3-Gallate

Epigallocatechin-3-gallate (EGCG) is the most abundant catechin derived from green tea. EGCG reduced the PSA levels, resulting in the suppression of their proliferation in LNCaP cells [25]. EGCG activated extracellular signal-regulated kinase (ERK1/2) and mitogen-activated protein kinase kinase (MEK) signaling, preventing the proliferation of PC-3 cells [23]. EGCG inhibited the invasion and migration of 22Rv1 cells via regulation of protein expression in VEGF, uPA, angiopoietin 1 and 2, MMP-2, and MMP-9 [24]. Combining EGCG and cisplatin promoted the expression of caspase 9, a pro-apoptotic splice isoform in PC-3 cells [26]. EGCG suppressed vasculogenic mimicry through inhibiting the Twist/VE-Cadherin/AKT pathway in PC-3 cells [27]. EGCG suppressed prostate cancer cell growth by modulating acetylation of androgen receptor [28]. EGCG antagonized bortezomib cytotoxicity by an autophagic mechanism in PC-3 cells [29].

Tea, especially green tea, has shown potential in preventing prostate cancer based on pre-clinical and preliminary clinical studies (reviewed by Henning et al. [148]). Our studies have also demonstrated that green tea and quercetin could significantly enhance the therapeutic effect of docetaxel in vitro and in mouse models at no increased risk of side effects [149,150]. However, the role of tea polyphenols in PCa still needs more long-term and intensive studies.

#### 2.1.8. Fisetin

Fisetin is a flavonol derived from *Acacia greggii*. Fisetin inhibited adhesion, migration, and metastasis by downregulating MMP-2 and MMP-9 and interfering with NF-κB signaling in PC-3 cells [30]. Fisetin sensitized the DU 145, LNCaP, and PC-3 cells to TRAIL-caused death by activating the receptor-mediated mitochondrial apoptotic pathways [33]. Fisetin also induced autophagic cell death by inhibiting mTOR and PI3K/Akt signaling [31]. Fisetin significantly inhibited proliferation, migration, and invasion by disrupting the microtubule dynamics in prostate cancer cells [34]. Moreover, Fisetin attenuated tumor growth and reduced serum PSA levels by competing with the AR ligand and decreasing AR stability in a CWR22 Rupsilon1 cell xenograft model in athymic nude mice [32].

#### 2.1.9. Formononetin

Formononetin (FN) is an O-methylated isoflavone isolated in *Trifolium pretense*. Formononetin-induced apoptosis was associated with the ERK/MAPK/Bax signaling in LNCaP and PC-3 cells [35]. Another study showed that formononetin provoked apoptosis by inhibiting IGF-1/IGF-1R signaling in PC-3 cells [36], modulating the Bax/Bcl-2 ratio, and altering the p38/Akt signaling [37]. Formononetin also triggers the mitochondrial apoptotic pathway, following the upregulation of RASD1 in DU 145 cells [38].

#### 2.1.10. Genistein

Genistein is a flavanone extracted from *Glycine max*. Genistein suppressed cell growth by inhibiting the IGF-1/IGF-1R signaling in PC-3 cells [39]. The genistein–gold nanoparticle conjugates have shown anti-proliferative activities on three malignant prostate carcinoma cell lines by MTT testing in vitro [151]. Genistein was reported to be essential for inhibiting PCa relapse and metastasis by targeting cancer stem cells (CSC) [152]. Genistein activated epigenetic modification of tumor suppression genes by reversing DNA hypermethylation, inhibiting cancer progression in DU 145 and PC-3 cells [153]. Genistein induced apoptosis by miR-1260b downregulating its target genes, sRRP1 and Smad4 [40]. Furthermore, genistein inhibited cell growth by regulating miR-34a and HOTAIR levels in PC-3 and DU 145 cells [41]. A low percentage of poorly differentiated tumors was observed in TRAMP mice fed genistein [154].

#### 2.1.11. Glycyrrhiza Compounds

The hexane/ethanol extract of *Glycyrrhiza uralensis* (HEGU) consists of the two active compounds, isoangustone A and licoricidin. Isoangustone A, an active flavonoid, induced apoptosis through increasing cleaved caspase-3, caspase-7, and caspase-9 in DU 145 cells [42]. In addition, isoangustone A diminished DNA synthesis by decreasing the CDK2 activity and caused G1 phase arrest by reducing the CDK2, CDK4, cyclin A, and cyclin D1 expression in DU 145 cells [43]. Licoricidin, another active flavonoid, inhibited the metastatic and invasive capacity of malignant PCa cells by suppressing the secretion of the matrix metalloproteinases (MMP-2, MMP-9), TIMP-1, urokinase-type plasminogen activator (uPA), and VEGF [44].

#### 2.1.12. Licochalcone

Licochalcone is an estrogenic flavonoid extracted from licorice root. Licochalcone caused G2/M phase arrest of PC-3 cells by suppressing cyclin B1 and cdc2 [45]. Licochalcone also induced autophagic cell death in LNCaP cells [155].

#### 2.1.13. Luteolin

Luteolin is a flavone extracted from *Terminalia chebula*. Luteolin suppressed angiogenesis and invasion by reducing AR, PSA, and VEGF-2R in LNCaP and PC-3 cells [47]. Luteolin acted as a ligand for the nuclear type II estradiol binding site, leading to epigenetic alterations in genes involved in the cell cycle in PC-3 cells [156]. Luteolin suppressed PC-3 cell xenografts by inhibiting IGF-1 and the subsequent activation of IGF-1R, AKT, EGFR, and MAPK/ERK signaling [46].

### 2.2. Polyphenol Lignans

#### 2.2.1. Arctigenin

Arctigenin is a phenylpropanoid dibenzyl butyrolactone found in the seeds of *Arctium lappa*. Our previous study showed that Arctigenin combined with quercetin synergistically enhanced the antiproliferative effect in prostate cancer cells [48]. Recently, we found that arctigenin inhibited prostate tumor growth in high-fat diet-fed mice through dual actions on adipose tissue and tumor [49]. Sun et al. have demonstrated that arctigenin may induce apoptosis and autophagy through the PI3K/Akt/mTOR pathway in PC-3M Cells [50].

#### 2.2.2. Honokiol

Honokiol is a lignan found in *Magnolia officinalis*. Honokiol caused G0-G1 phase arrest by triggering apoptotic DNA fragmentation in LNCaP, PC-3, and C4-2 cells [51]. Likewise, honokiol exhibited growth inhibitory effects by the pro-apoptotic and anti-angiogenic mechanism on PCa xenografts [52].

#### 2.2.3. Magnolol

Magnolol is a hydroxylated biphenyl lignan extracted from the root and stem bark of *Magnolia officinalis*. Magnolol induced apoptotic cell death through epidermal growth factor receptor (EGFR)-mediated pathways and inhibited the adhesion, invasion, and migration in PC-3 cells [53].

#### 2.2.4. Obovatol

Obovatol, a biphenyl ether lignan found from *Magnolia obovate*. Obovatol engaged LNCaP and PC-3 cells to apoptotic cell death by inhibiting NF-κB activity, thereby enhancing the therapeutics’ inhibitory effect on PCa cell growth [54].

#### 2.2.5. Silibinin

Silibinin or silybin is a flavonolignan derived from the fruits of *Silybum marianum*. Silibinin induced G1 phase arrest by reducing the expression of p21 and p27 in DU 145 cells [55]. Silibinin induced apoptosis by inhibiting active Stat3 and restraining Wnt/LRp6 signaling, while it sensitized DU 145 cells to TNFα-induced apoptosis by inactivating NF-κB [56]. Silibinin inhibited the epithelial to mesenchymal transition (EMT) of PCa cells by interfering with the NF-κB signaling and subsequent reduction in ZEB1 and SLUG transcription factors [57]. Silibinin also prevented PCa cells-induced osteoclastogenesis in a high bone metastatic prostate model [157]. Silibinin’s inhibitory effects on PC-3 cells xenografts were attributed to increased expression of IGFBP-3, Cip1/p21, and Kip1/p27, and a reduced expression of Bcl-2 and VEGF [158]. Silibinin confined tumor microvessel density by a reduction in VEGF, VEGFR-2, MMPs, and vimentin, thus blocking PCa growth and progression in TRAMP mice [58].

### 2.3. Polyphenol Stilbenes

#### Resveratrol

Resveratrol, a natural stilbenoid polyphenol, is present in several fruits, such as tomatoes, grapes, blueberries, mulberries, and raspberries. Resveratrol increased the pro-apoptotic potential of TNF-related apoptosis-inducing ligand (TRAIL) by activating FKHRL1 [59]. Resveratrol reduced the metastatic potential of PCa cells by decreasing the expression of vascular endothelial growth factor (VEGF), VEGF receptor 2 (VEGFR2), and matrix metalloproteinases (MMPs) [60]. Resveratrol downregulated AR expression and altered the chemokine receptor type 4 (CXCR4) signaling in TRAMP mouse models [61]. Resveratrol enhanced cellular antioxidant defense ability by reducing ROS, reactive nitrogen species (RNS), and inducing the antioxidant enzyme heme-oxygenase-1 (HO-1) [62]. Resveratrol inhibited hypoxia-inducible factor (HIF)-1α and β-catenin-mediated AR signaling in CRPC [63]. These results merit further studies of resveratrol for its potential efficacy against PCa.

### 2.4. Other Polyphenols

#### 2.4.1. Curcumin

Curcumin is a polyphenolic compound extracted from the rhizome of the plant *Curcuma longa*. Curcumin was first described to induce apoptosis of PCa cells by interfering with the EGF-R signaling [64]. Curcumin induced apoptosis through apoptosis-inducing factor (AIF) and caspase-independent pathway in PC-3 cells [65]. Curcumin blocked the occupancy at sites of AR function by interacting with cAMP response element binding protein (CBP) and co-activator protein p300 in LNCaP and PC-3 cells, thus decreasing tumor growth and delaying the onset of hormone-resistant disease [159]. Curcumin affected Wnt/β-catenin signaling, resulting in autophagy in early-stage PCa [66]. Curcumin was also an ideal candidate for treating PCa by activating the Nrf-2 signaling [67]. Additionally, derivates of curcumin enhanced expressions of Nrf-2 and phase II detoxifying genes by epigenetic regulation in TRAMP C1 PCa cells [160]. Curcumin reduced the PCa growth and metastasis rate in LNCaP xenograft models [69]. Curcumin suppressed the invasion and metastasis of DU 145 xenografts by a reduction in metalloproteinase expression [68].

#### 2.4.2. Ellagitannin

Ellagitannins, belonging to polyphenols, are enriched in the pomegranate fruit’s seeds and juice. Punicalagin, a kind of ellagitannin, is broken down to ellagic acid (EA), and then further metabolized to urolithin A at intestinal pH by gut microbiota. EA modulated AIF expression, leading to ROS generation and caspase-mediated apoptosis in LNCaP cells [70]. EA decreased the eicosanoid biosynthesis levels in LNCaP cells, depicting the anti-angiogenetic effects [71]. EA could induce apoptosis and cell cycle arrest in the S phase by decreasing cyclin B1 and D1 expression in a caspase-dependent pathway in DU 145 and PC-3 cells [72]. Moreover, EA interfered with protease activity and reduced the MMP-2 secretion, confining the invasive potential of PC-3 and rat PCa cells [73]. The presence of urolithin A led to the inhibition of MDM2-mediated p53 polyubiquitination in 22RV1 and PC-3 cells, indicating that urolithin A inhibited PCa via the p53-MDM2 signaling [74]. In addition, the metabolites ellagic acid, luteolin, and punicic acid have shown the capacity to inhibit PCa cell metastasis and angiogenesis in murine studies [75].

#### 2.4.3. Gallic Acid

Gallic acid (GA) is a polyphenolic constituent extracted from grape seeds. GA caused DNA damage through increasing cdc25A/C-cdc2 phosphorylation, resulting in growth inhibition and G2/M cell cycle arrest in DU 145 cells [76]. GA triggered mitochondrial-mediated mechanisms, including ROS generation, the cleavage of caspase-3, caspase-9, and poly (ADP-ribose) polymerase (PARP), inducing PCa cells apoptosis [77]. GA blocked the p38, JNK, PKC, and PI3K/AKT pathways, inhibiting the invasion and migration of PC-3 cells [78]. The oral feeding with water supplemented with GA decreased the expression of Cdk2, Cdk4, Cdk6, cyclin B1, and E proteins, inhibiting PCa progression to advanced-stage adenocarcinoma in TRAMP mice [79]. GA reduced the microvessel density of DU 145 and 22Rv1 xenografts, suppressing tumor growth in nude mice [80].

#### 2.4.4. Gossypol

Gossypol is a polyphenolic aldehyde extracted from cottonseed. Gossypol modulated the expression of cyclin D1, Cdk4, and phospho-Rb through TGF-β1 and Akt signaling, leading to G0/G1 phase arrest in MAT-LyLu cells [81]. Gossypol inhibited the heterodimerization of Bcl-xL/Bcl-2 with pro-apoptosis molecules, resulting in apoptotic processes [82]. In addition, gossypol induced autophagy by releasing the BH3-only pro-autophagic protein Beclin1, which, in turn, triggered the autophagic cascade in androgen-independent PCa cells [83]. Gossypol inhibited metastatic behaviors and angiogenesis by repressing NF-κB and AP-1 activity in PC-3 cells [84]. Gossypol suppressed the phosphorylation of focal adhesion kinase, extracellular signal-related kinase, and key intracellular proangiogenic kinases by blocking VEGF receptor 2 kinase activation, causing the subsequent suppression of angiogenesis in PC-3 xenografts [85].

## 3. Terpenoids

### 3.1. Artemisinin

Artemisinin, a potent anti-malarial compound, is a sesquiterpene lactone isolated from *Artemisia annua*. Artemisinin transcriptionally downregulated the CDK4 expression by disrupting Sp1 interactions with the CDK4 promoter, triggering G1 cell cycle arrest in LNCaP cells [86]. Dihydroartemisinin (DHA), a derivative of artemisinin, has been shown to reduce cell viability by activating caspases 8/9 in DU 145, LNCaP, and PC-3 cells [87].

### 3.2. Betulinic Acid

Betulinic acid (BA) is a pentacyclic triterpene extracted from the bark of *Betula papyrifera*. BA shifted the Bax/Bcl-2 ratio and cleaved PARP by inhibition of NF-κB, thus inducing PC-3 cell apoptosis [88]. BA decreased oncoproteins expression by inhibiting multiple deubiquitinases (DUBs), which resulted in poly-ubiquitinated protein accumulation, thus increasing apoptosis in DU 145, LNCaP, and PC-3 cells [89]. Furthermore, BA inhibited angiogenesis and tumor growth by decreasing the expression of AR and cyclin D in TRAMP mice [90].

### 3.3. Germacrone

Germacrone is a sesquiterpene that has been isolated from *Geranium macrorrhizum*. Yu et al. have shown that Germacrone induces apoptosis and protective autophagy in human prostate cancer cells [91].

### 3.4. Ginsenosides

Ginsenosides are a class of natural steroid glycosides and triterpene saponins extracted exclusively from the genus *Panax*. Ginsenoside Rg3, one of the bioactive components extracted from ginseng root, downregulated aquaporin 1 (AQP1) expression by interfering with the p38 MAPK signaling, inhibiting migration and metastasis of PC-3M cells [92]. Rh2 ginsenoside, another bioactive glycoside, modulated MAP kinase expression, causing cell detachment and showing an anti-proliferative effect in LNCaP and PC-3 cells [93]. Oral administration of Rh2 significantly increased the apoptosis rate, eventually reducing the tumor growth in a PC-3 xenograft model [94].

### 3.5. Glycyrrhizin

Glycyrrhizin, also known as liquorice, is extracted from the root of a flowering plant of the bean family, Fabaceae. Gioti et al. have demonstrated liquorice’s anti-proliferative properties by both apoptosis and autophagy mechanisms in PC-3 cells [95].

### 3.6. Lycopene

Lycopene is the most abundant antioxidant carotenoid that colors tomatoes. It was reported that lycopene inhibited PCa cell proliferation via modulation of CDK7, epidermal growth factor receptor (EGFR), insulin-like growth factor-1 receptor (IGF-1R), and BCL2 [96]. Lycopene reduced cholesterol synthesis by activating the PPARγ–LXRα–ABCA1 axis in LNCaP and DU 145 cells [97]. Increasing concentrations of lycopene induced the release of mitochondrial cytochrome c, and reduced mitochondrial potential in LNCaP cells [161]. A high concentration of lycopene induced apoptosis through alterations in IGF-I, IGF-IR, and IGFBP-3 expression in PC-3 cells and xenograft models [98]. Finally, the therapeutic activity of lycopene was also observed in TRAMP mice fed lycopene [99].

### 3.7. Oridonin

Oridonin is an isoprenoid extracted from *Rabdosia rubescens*. Oridonin elicited apoptosis and G0/G1 cell cycle arrest by upregulating p53 and Bax and downregulating Bcl-2 in LNCaP cells [100]. Oridonin triggered apoptosis, autophagy, and G2/M phase arrest by the up-regulation of p21 in LNCaP and PC-3 cells [101].

### 3.8. Thymoquinone

Thymoquinone (TQ) is a monoterpene extracted from *Nigella sativa*. TQ inhibited cell growth by decreasing AR, E2F-1, and E2F-1 in LNCaP, PC-3, C4-B, and DU 145 cells [102]. TQ induced apoptosis through increasing ROS generation and decreasing GSH levels in C4-B and PC-3 cells [103]. Furthermore, TQ was demonstrated to prevent tumor angiogenesis by suppressing AKT in PC-3 cells [104].

### 3.9. Ursolic Acid

Ursolic acid (UA) is a pentacyclic triterpenoid compound isolated from *Cornus officinalis*. UA evoked apoptosis and confined invasion by inhibiting Akt and MMP9 in PC-3 cells [105]. UA also induced apoptosis through JNK activation, resulting in phosphorylation and degradation of Bcl-2 and activation of caspase 9 in LNCaP, LNCaP-AI, and DU 145 cells [106]. UA restricted metastasis by repressing CXCR4 expression in PCa cells [107]. In addition, UA induced apoptosis by downregulating the NF-κB and STAT3 target genes involved in proliferation, survival, and angiogenesis in TRAMP mice [108].

### 3.10. β-Elemonic Acid

*β*-Elemonic acid (β-EA), a known triterpene isolated from *Ganoderma tsugae*, *Ganoderma lucidum*, and *Boswellia*, exhibits anti-inflammatory effects. Bao et al. have shown that β-EA inhibited the activation of JAK2/STAT3/MCL-1 and the NF-κB pathway in PCa cells. Furthermore, tumor growth in a murine xenograft model was retarded by the administration of β-EA [109].

## 4. Taxanes

Taxane is a class of diterpenes, initially isolated from the genus *Taxus* (yews). Therapies aimed at decreasing androgen signaling have been the principal treatment for advanced prostate cancer for most of a century [162]. The breakthrough in cytotoxic chemotherapy for prostate cancer came with the taxanes. Taxanes function through microtubule interaction and polymerization, stabilizing microtubule stabilization and mitotic arrest.

### 4.1. Cabazitaxel

Cabazitaxel is a semi-synthetic derivative of a natural taxoid extracted from yew needles. Cabazitaxel is a third-generation taxane drug developed after resistance was seen with the other taxanes. Cabazitaxel effectively inhibited the proliferation of CRPC cells with acquired docetaxel resistance [163,164]. The distinct activity of cabazitaxel to overcome resistance to prior taxanes or hormonal therapies in prostate cancer was observed in pre-clinical and clinical data [110]. In conclusion, the success of taxanes in PCa treatment suggests a need for a further understanding of the taxane mechanisms of action and a better development of rational taxane-based combination therapies.

### 4.2. Docetaxel (DTX)

Docetaxel is a taxane derivative isolated from the European yew tree’s renewable and more readily available leaves. Docetaxel is a semi-synthetic analog of paclitaxel as a cytotoxic chemotherapeutic agent. Gan et al. have shown that silencing the p38/p53/p21 signaling could be crucial to sensitizing LNCaP cells to docetaxel treatment [111].

### 4.3. Paclitaxel (PTX)

Paclitaxel is a chemotherapy medication in the taxane family of drugs, first isolated from the Pacific yew, *Taxus brevifolia*. Paclitaxel phosphorylated the B-cell lym phoma-2 (Bcl-2) protein at serine residues, which inhibited the mitochondrial release of cytochrome c and subsequently blocked the caspase cascade, thereby inducing apoptosis [112]. Paclitaxel induced nuclear accumulation of FOXO1, thus inhibiting AR nuclear translocation from reducing the expression of AR target genes in PCa cells [113].

## 5. Alkaloids

### 5.1. Anibamine

Anibamine, a novel pyridine quaternary alkaloid recently isolated from *Aniba* spp., is a natural antagonist of C–C chemokine receptor type 5 (CCR5). Haney et al. found anibamine and their analogs, with micromolar range affinity to CCR5, exerted anti-proliferative activity against several prostate cancer cell lines [114].

### 5.2. Berberine

Berberine is an isoquinoline alkaloid isolated from the genus *Berberis*. Low concentrations of berberine triggered G1 arrest by activating the p53-p21 cascade in RM-1 cells [115]. Berberine inhibited tumor growth by reducing AR expression in nude mice bearing LNCaP xenografts [116]. Berberine inhibited the expression of HIF-1 alpha and VEGF by interfering with the MAPK/caspase-3 and ROS pathways, thereby enhancing the radiosensitivity of human PCa cells [117].

### 5.3. Capsaicin

Capsaicin is a vanilloid extracted from red pepper. Capsaicin induced apoptosis through ROS generation and activation of caspase 3 in PC-3 cells [165]. Further studies revealed that capsaicin induced apoptosis by ceramide accumulation and activation of JNK and ERK in PC-3 cells [118]. Likewise, the various concentrations of capsaicin stimulated apoptosis by increasing p53, p21, and Bax, decreasing prostate-specific antigen (PSA) and AR, and inhibiting proteasome activity in LNCaP and DU 145 cells [119].

### 5.4. Neferine

Neferine is the major bisbenzylisoquinoline alkaloid isolated from the seed embryo of a traditional medicinal plant, *Nelumbo nucifera* (Lotus). Erdogan et al. have shown that neferine inhibits the proliferation and migration of human prostate cancer stem cells through p38 MAPK/JNK activation [120].

### 5.5. Piperine

Piperine is a major bioactive alkaloid present in black pepper. Piperine induced apoptosis, promoted autophagy, triggered cell cycle arrest at G0/G1, and inhibited proliferation in PCa cells and animal xeno-transplanted models [122]. Piperine also inhibited the transcription factor NF-kB expression and downregulate phosphorylated STAT-3 in LNCaP, PC-3, and DU 145 cells [121]. Additionally, piperine remarkably enhanced the anti-cancer effects of docetaxel in a xenograft model of CRPC [123].

### 5.6. Sanguinarine

Sanguinarine is a benzophenanthridine alkaloid isolated from *Sanguinaria canadensis*. Sanguinarine restricted PCa cell growth, migration, and invasion by inactivating Stat3 in DU 145, C4-2B, and LNCaP cells [124]. The administration of sanguinarine confined tumor weight and volume by suppressing survivin via the ubiquitin–proteasome system in DU 145 cell xenografts [125]. Using a high-throughput screen, Bodle et al. discovered two natural compounds, sanguinarine and celastrol, to be cytotoxic against prostate cancer cell lines by inhibiting interaction with regulator of G protein signaling 17 (RGS17) [126].

## 6. Other

### 6.1. Gambogic Acid

Gambogic acid (GA) is a xanthone found in *Garcinia hanburyi*. GA suppressed TNF-α-caused invasion of PC-3 cells by inactivating PI3K/Akt and NF-κB pathways [127]. Injection of GA suppressed angiogenesis and tumor growth by inhibiting vascular endothelial growth factor receptor 2 (VEGF-2R) and its downstream protein kinases, such as c-Src and AKT, in a xenograft model [128].

### 6.2. Glucoraphanin

Glucoraphanin (GRA) is the most abundant glucosinolate (GSLs) and is extracted from cruciferous vegetables such as brussels sprouts, broccoli, and cauliflower, contributing to these vegetables’ organoleptic properties. GSLs are enzymatically hydrolyzed to breakdown derivates, including thiocyanates, isothiocyanates, and indoles, exhibiting biological effects (Figure 1), either by bacteria with microbial thioglucosidase activity in the gut or by the plant enzyme myrosinase [166].

#### 6.2.1. Sulforaphane

Sulforaphane (SFN) was first isolated from *Brassica oleracea* and is a natural isothiocyanate (ITC). SFN could reduce the progression and incidence of PCa by metabolic regulation in PCa cells [167]. SFN interfered with heat shock protein 90 (Hsp90), a critical androgen receptor (AR) chaperone, in prostate cancer [130]. SFN directly attenuated the androgen receptor (AR) pathways by inhibiting HDAC6 activity in the LNCaP cells [131]. SFN metabolites, sulforaphane-cysteine (SFN-Cys), and sulforaphane-N-acetyl-cysteine (SFN-NAC) disrupted the microtubules and induced apoptosis in DU 145 and PC-3 cells by phosphorylation of extracellular signal-regulated protein kinases 1 and 2 (ERK1/2) [129]. SFN could cause G2/M cell cycle arrest and apoptotic cell death via signaling disruption within tumor microenvironments [168]. SFN has been reported to strongly inhibit the nuclear translocation of p65 and nuclear factor-kappa B (NF-ĸb) activity, subsequently regulating gene expression of Bcl-XL, cyclin D1, and VEGF in PCa cells [169]. SFN also restricts prostate cancer cell migration because of the modulations of the Notch pathway [170]. SFN induced apoptosis by reactive oxygen species (ROS) generation, triggering intrinsic and extrinsic caspase cascades in LNCaP, PC-3, and DU 145 cells [132]. SFN significantly reduced the level of free fatty acid (FA) and relevant metabolism proteins in PCa cells and the prostatic adenocarcinoma of TRAMP mice, suggesting that the SFN inhibits PCa progression by reducing FA metabolism [133]. In addition, SFN inhibited HDAC3, an epigenetic regulator, in PC-3 cells and decreased HDAC3 expression in TRAMP mice [134]. These results explain the mechanism of how the cruciferous vegetable broccoli may reduce the risk of PCa progression.

#### 6.2.2. Phenethyl-Isothiocyanate

Phenethyl-isothiocyanate (PEITC) is another extensively studied ITC found in cruciferous vegetables. PEITC induced G2/M cell cycle arrest by inhibiting the expression of α- and β-tubulin and generation of ROS in C4-2B, DU 145, PC-3, and LNCaP cells [135]. PEITC activated Bax and ROS production to trigger apoptotic mechanisms in LNCaP and PC-3 cells [136]. It was noteworthy that PEITC interfered with the Notch pathway and inactivated Akt with subsequent suppression of VEGF, restraining the migration of PC-3 and LNCaP cells [137]. PEITC caused apoptosis and cell cycle arrest in the G2/M phase in a dose-dependent manner in DU 145 cells [171]. Finally, PEITC slowed tumor growth by suppressing angiogenesis in an LNCaP xenograft model [138]. Studies in TRAMP mice also revealed that the overexpression of E-cadherin-induced prostate carcinogenesis was inhibited by PEITC [139].

#### 6.2.3. Indole-3-Carbinol

Indole-3-carbinol is a main breakdown product of GRA by the action of myrosinase enzymes. Indole-3-carbinol caused cell cycle arrest by downregulating cyclin-dependent kinase2 (CDK2), CDK4, and CDK6, and upregulating p15, p21, and p27 [140]. Indole-3-carbinol induces apoptosis by activating cas-3 and cas-9, the subsequent release of mitochondrial cytochrome C, and upregulation of the pro-apoptotic protein Bax [140]. Indole-3-carbinol potentiated the effects of tumor necrosis factor-related apoptosis-inducing ligand (TRAIL) by inactivation of NF-ĸB, estrogen receptor (ER), androgen receptor (AR), and nuclear factor-E2-related factor 2 (Nrf2) signaling [140]. Additionally, indole-3-carbinol could modulate epigenetic alterations of cancer stem cells, including aberrant microRNA expression, CpG methylation, and histone modification [168].

## 7. Conclusions

This review represents the latest findings on phytochemicals from vegetables, fruit, tea, and herbs that have shown multi-modal anti-cancer effects on PCa cells, including cell cycle arrest, triggering apoptotic cell death, and suppression of invasion/metastasis, and their underlying molecular mechanisms (Figure 2). The multi-targeting capacity of these phytochemicals seems to be a favorable feature against cancer cells that typically have lots of dysregulated signals that may crosstalk with each other. Moreover, many phytochemicals have been selective in targeting cancer cells without toxicity to normal cells, making them more promising as cancer preventive/therapeutic agents. In addition, increasing evidence from laboratory studies has shown the potential of phytochemicals to enhance the therapeutic effect of drugs in a non-/less-toxic manner in treating different cancers, including prostate cancer.

However, several limitations with these phytochemicals need to be addressed on the road to a successful clinical application. First, the low bioavailability is one of the major limitations of most phytochemicals. In vitro effective concentrations are barely achievable in vivo through oral consumption at safe doses, limiting their clinical success. Second, the long-term toxicity of these phytochemicals will still need to be tested, particularly due to the relatively high concentrations consumed during disease treatment. In addition, complete profiling of their molecular targets may help address the concern of possible side effects of their multi-targeting activities.

Relative to the extensive pre-clinical studies on phytochemicals, information from clinical trials is limited. All phases of clinical trial studies are needed to advance this field. In addition, efforts should continue to identify novel phytochemicals of improved bioavailability and anti-cancer potency. Despite all the challenges, natural products remain a promise to be an alternative or complementary approach to cancer treatment. In conclusion, elucidation of the chemical structure-based precise mechanisms of action will facilitate the identification of novel natural compounds with significant anti-cancer properties, for both drug development and optimization of drug combinations, which are essential to make the phytochemical an anti-cancer agent in a non-/less-toxic manner in the treatment of prostate cancer, thereby providing significant health benefits for humans.

## Figures and Tables

**Figure 1 biomolecules-12-01306-f001:**
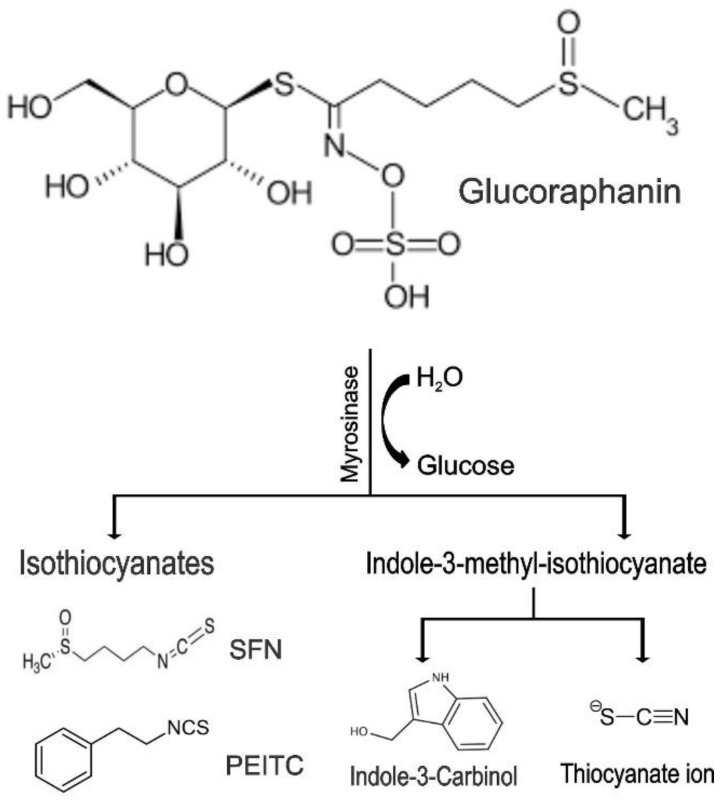
Breakdown of glucosinolates.

**Figure 2 biomolecules-12-01306-f002:**
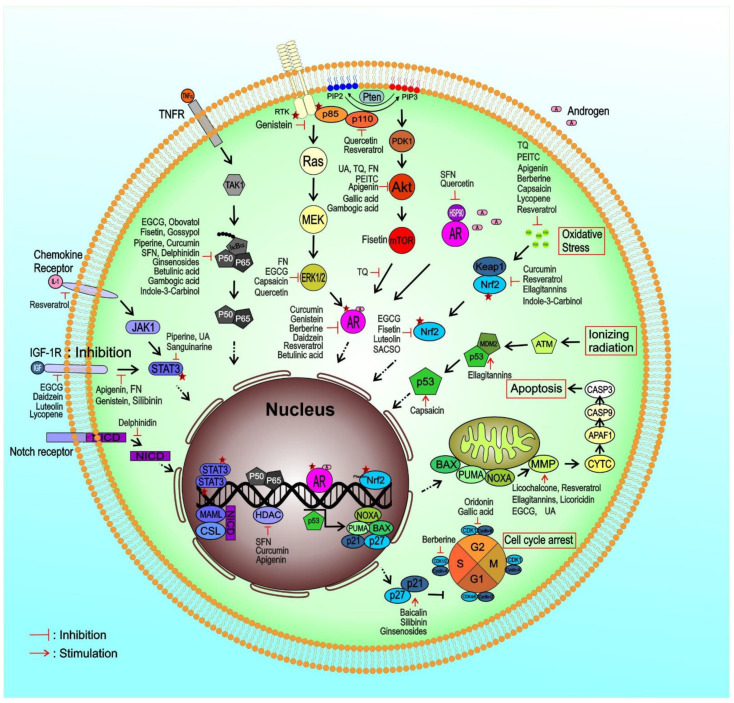
Mechanism of action of bioactive natural products in prostate cancer cells.

**Table 1 biomolecules-12-01306-t001:** Bioactive natural products against prostate cancer.

Natural Compound	Constituent	Plant Source	Chemical Structure	Mechanism	Molecular Pathway (Reference)	In Vitro	In Vivo
1. Polyphenol
1. 1 Polyphenol Flavonols
1.1.1 Quercetin	Flavonol	Apple and onion	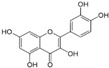	Induction of apoptosis.	PI3K/Akt and MAPK/ERK [6]; p21, FA, Hsp90 [7].	LNCaP, DU 145, PC-3, Prostate cancer stem cells.	
1.1.2 Apigenin	Flavone	Anthemis sp.	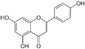	Induction of apoptosis and cell cycly arrest.	IGF-IR; PI3k/Akt [8]; HDACs [9]; ROS [10]; VEGF [11].	PC-3 and 22Rv1.	22Rv1 and PC-3 xenografts.
1.1.3 Baicalin	Flavone	Genus Scutellaria	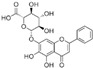	Inhibition of proliferation and induction of apoptosis.	Caveolin-1/AKT/mTOR [12]; AR target genes [13]; Ezrin [14].	DU 145 and PC-3.	PC-3 xenografts.
1.1.4 Cyanidin-3-glucoside	Flavonoid anthocyanin	Berries, red cabbages, teas, and coloured grains.	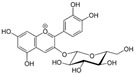	Inhibition of EMT.	Snail/E-cadherin [15].	PC-3.	
1.1.5 Daidzein	Isoflavone	Soybeans	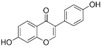	Induction of cell cycle arrest; CpG island demethylation.	CDK [16]; GSTP1 and EPHB2 [17]; ERβ [18].	DU 145, LNCaP and PC-3.	
1.1.6 Delphinidin	Flavonoid anthocyanin	Viola sp.	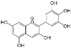	Inhibition of cell growth; Induction of apoptosis.	NF-κB [19]; β-catenin [20]; DR5 [21]; HDAC [22].	PC-3 and LNCaP.	PC-3 xenografts.
1.1.7 Epigallocatechin-3-gallate (EGCG)	Catechin	Green tea	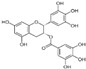	Suppression of the proliferation, invasion and migration, and vasculogenic mimicry; Induction of apoptosis and autophagy.	ERK1/2 [23], VEGF, uPA, angiopoietin 1/2, MMP-2, and MMP-9 [24], PSA [25], Caspase 9 [26]; Twist/VE-Cadherin/AKT [27]; AR [28]; CHOP and p-eIF2α [29].	LNCaP, PC-3, 22Rv1.	PC-3 xenografts.
1.1.8 Fisetin	Flavonol	Acacia greggii	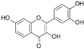	Inhibition of adhesion, migration, and metastasis; Induction of autophagy.	NF-κB [30]; mTOR and PI3K/Akt [31]; AR [32]; NF-κB [33]; Nudc [34].	DU 145, LNCaP, PC-3.	CWR22 Rupsilon1 cells xenograft.
1.1.9 Formononetin (FN)	Omethylated isoflavone	Trifolium pretense.	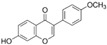	Induction of apoptosis.	ERK1/2, MAPK-Bax [35]; IGF-1/IGF-1R [36]; p38/Akt [37]; RASD1 [38].	LNCaP, PC-3, and DU 145.	
1.1.10 Genistein	Flavanone	Glycine max	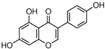	Inhibition of the proliferation;Suppression of cancer stem cells; Activation of epigenetic modification; Induction of apoptosis.	IGF-1/IGF-1R [39]; miR-1260b [40]; miR-34a and HOTAIR [41].	LNCaP, PC-3, and DU 145.	TRAMP mice.
1.1.11 Glycyrrhiza
1.1.11.1 Isoangustone A	Flavonoid	Glycyrrhiza uralensis	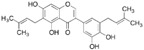	Induction of apoptosis; G1 phase arrest.	Caspase [42]; CDK2/4, cyclin A [43].	DU 145.	
1.1.11.2 Licoricidin	Flavonoid	Glycyrrhiza uralensis	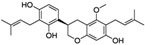	Inhibition of metastasis and invasion.	MMP, TIMP-1, VEGF [44].	DU 145.	
1.1.12 Licochalcone	Flavonoid	Licorice root	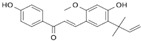	Induction of G2/M phase arrest and apoptosis.	Cyclin B1 and cdc2 [45].	LNCaP, PC-3.	
1.1.13 Luteolin	Flavone	Terminalia chebula	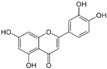	Suppression of angiogenesis; Induction of cell cycle arrest.	IGF-1 [46]; VEGF-2R, AR [47].	LNCaP and PC-3.	PC-3 xenografts.
**1.2 Polyphenol lignans**
1.2.1 Arctigenin	lignan	Arctium lappa	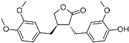	Inhibition of the proliferation; Induction of apoptosis and autophagy.	AR [48], FFA [49], PI3K/Akt/mTOR [50].	*LAPC-4 and LNCaP.*	**LAPC-4 xenograft.**
1.2.2 Honokiol	Lignin	Magnolia officinalis	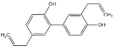	Induction of G0-G1 phase arrest and apoptosis; Inhibition of angiogenesis.	Bax, Bak, Bad, Bcl-xL, and Mcl-1 [51,52].	LNCaP, PC-3, and C4-2.	PC-3 xenografts.
1.2.3 Magnolol	Hydroxylated biphenyl lignan	Magnolia officinalis	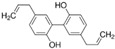	Induction of apoptosis and inhibition of the adhesion, invasion, and migration.	EGFR [53].	PC-3.	
1.2.4 Obovatol	Biphenyl ether lignan	Magnolia obovate	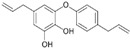	Induction of apoptosis.	NF-κB [54].	LNCaP and PC-3.	
1.2.5 Silibinin	Flavolignan	Silybum marianum	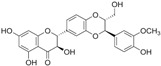	Induction of G1 phase arrest; Induction of apoptosis; Inhibition of EMT; Restriction of tumor microvessel density.	p21 and p27 [55]; NF-κB [56]; ZEB1 and SLUG [57].	DU 145.	PC-3 xenografts and TRAMP mice [58].
**1.3 Polyphenol Stilbenoids**
1.3.1 Resveratrol	Stilbenoid	Grape, raspberry, mulberry.		Induction of apoptosis; Cellular antioxidant defense.	FKHRL1 [59]; VEGF, MMPs [60]; CXCR4 [61]; ROS, RNS, HO-1 [62]; AR [63].		TRAMP mouse [61].
**1.4 Other Polyphenols**
1.4.1 Curcumin	Polyphenols	Curcuma longa	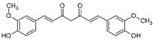	Induction of apoptosis; Induction of autophagy.	EGF-R [64]; AIF [65]; Wnt/β-catenin [66]; Nrf-2 [67]; MMPs [68].	DU 145, LNCaP and PC-3.	LNCaP xenograft [69]; DU 145 xenografts [68].
1.4.2 Ellagitannins	Polyphenols	Pomegranate fruit	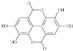	Induction of apoptosis and S phase cell cycle arrest; Inhibition of metastasis and angiogenesis.	SIRT1, p21, AIF [70]; Eicosanoid [71]; cyclin B1 and D1 [72]. MMP-2 [73]; p53-MDM2 [74].	LNCaP, 22RV1 and PC-3.	murine studies [75].
1.4.3 Gallic acid	Polyphenols	Grape seed	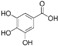	Induction of G2/M cell cycle arrest; Induction of apoptosis; Inhibition of invasion and migration.	cdc25A/C-cdc2 [76]; PARP [77]; p38, JNK, PKC, and PI3K/AKT [78]; Cdk, cyclin B1, and E [79].	DU 145; PC-3.	TRAMP mice [79]; DU 145 and 22Rv1 xenografts [80].
1.4.4 Gossypol	Polyphenolic aldehyde	Cotton seed	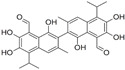	Induction of G0/G1 phase arrest; Induction of apoptosis; Induction of autophagy; Inhibition of angiogenesis; Reduction of the microvessel density.	TGF-β1 and Akt [81]; Bcl-xL [82]; Beclin1 [83]. AP-1, NF-κB [84].	MAT-LyLu; PC-3.	PC-3 xenografts [85].
**2. Terpenoids**
2.1 Artemisinin	Sesquiterpene	Artemisia annua	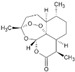	Induction of G1 cell cycle arrest; Inhibition of viability.	CDK4 and Sp1 [86], caspases 8/9 [87].	LNCaP, DU 145 and PC-3.	
2.2 Betulinic Acid (BA)	Triterpene	Betula papyrifera		Induction of apoptosis; Inhibition of angiogenesis.	Bax/Bcl-2 [88], DUBs [89], AR and cyclin D [90].	DU 145, LNCaP, and PC-3.	TRAMP mice [90].
2.3 Germacrone	Sesquiterpene	Geranium macrorrhizum.	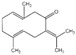	Induction of apoptosis and autophagy.	Akt/mTOR [91].	PC-3 and 22RV1.	
2.4 Ginsenosides	Steroid glycosides and triterpene saponins	Genus Panax	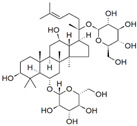	Inhibition of proliferation; Inhibition of migration and metastasis; Induction of apoptosis.	p38 [92]; MAP [93].	PC-3M, LNCaP and PC-3.	PC-3 xenograft [94].
2.5 Glycyrrhizin	Saponins	Fabaceae	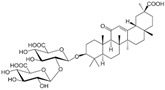	Induction of apoptosis and autophagy.	Apoptosis and autophagy [95].	PC-3.	
2.6 Lycopene	Carotenoid	Tomatoes	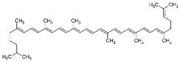	Reduction in cholesterol synthesis; Induction of apoptosis.	CDK7, EGFR, IGF-1R, and BCL2 [96]; PPARγ-LXRα-ABCA1 [97]; CDK7,EGFR, IGF-1R, BCL2 [96]; IGF-I, IGF-IR, and IGFBP-3 [98].	LNCaP, PC-3 and DU 145.	PC-3 xenograft [98]; TRAMP mice [99].
2.7 Oridonin	Isoprenoid	Rabdosia rubescens	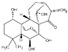	Induction of apoptosis and G0/G1 cell cycle arrest; Induction of apoptosis, autophagy, and G2/M phase arrest.	p53 and Bax [100]; p21 [101].	LNCaP and PC-3.	
2.8 Thymoquinone (TQ)	Monoterpene	Nigella sativa	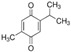	Induction of apoptosis; Inhibition of angiogenesis.	AR and E2F-1 [102]; ROS [103]; AKT, VEGF [104].	LNCaP, PC-3, C4-B, and DU 145.	
2.9 Ursolic acid (UA)	Pentacyclic triterpenoid	Cornus Officinalis	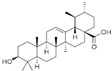	Induction of apoptosis.	Akt and MMP9 [105]; JNK [106]; CXCR4 [107]. NF-κB and STAT3 [108];	PC-3, LNCaP, LNCaP-AI, and DU 145.	TRAMP mice [108].
2.10 β-Elemonic acid	Triterpene	Ganoderma tsugae, lucidum, and Boswellia.	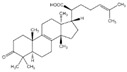	Induction of apoptosis.	JAK2/STAT3/MCL-1 and NF-κB [109].	22RV1.	22RV1xenograft [109].
**3. Taxanes**
3.1 Cabazitaxel	Taxane	European yew tree	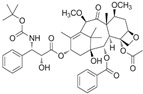	Inhibition of proliferation.	P-gp [110].	C4-2.	
3.2 Docetaxel	Taxane	European yew tree	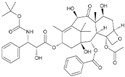	Induction of apoptosis.	p38/p53/p21 [111].	LNCaP, PC3 and DU 145.	
3.3 Paclitaxel	Taxane	Pacific yew tree	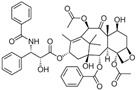	Induction of apoptosis.	Bcl-2 [112]; AR [113].	C4-2.	22Rv1 xenografts [113].
**4. Alkaloids**
4.1 Anibamine,	Pyridine quaternary alkaloid	Aniba sp.	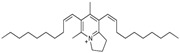	Inhibition of proliferation.	CCR5 [114].	LNCaP.	
4.2 Berberine	Isoquinoline Alkaloid	Genus Berberis	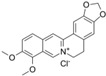	Induction of G1 phase arrest; enhancement of the radiosensitivity.	p53-p21 [115], AR [116], MAPK/caspase-3 and ROS [117].	RM-1.	LNCaP xenografts [116].
4.3 Capsaicin	Alkaloid	Red pepper	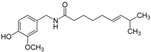	Induction of apoptosis.	JNK and ERK [118]. p53, p21, and Bax, AR [119].	LNCaP, PC-3, and DU 145.	
4.4 Neferine	Bisbenzylisoquinoline alkaloid	Nelumbo nucifera	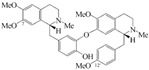	Inhibition of proliferation and migration of prostate cancer stem cells.	p38 mapk/jnk [120].	PC3.	
4.5 Piperine	Alkaloid	Black pepper	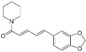	Induction of apoptosis, autophagy, and G0/G1 phase arrest.	NF-kB, STAT-3 [121].	LNCaP, PC-3, and DU 145.	Animal xeno-transplanted model [122]; CRPC xenograft model [123].
4.6 Sanguinarine	Alkaloid	Sanguinaria Canadensis	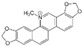	Inhibition of ubiquitin-proteasome system.	Stat3 [124], survivin [125], RGS17 [126].	DU 145, C4-2B, and LNCaP.	DU 145 xenografts [125].
**5. Other**
5.1 Gambogic acid	Xanthone	Garcinia hanburyi	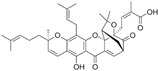	Inhibition of angiogenesis.	PI3K/Akt and NF-κB [127]; VEGF-2R [128].	PC-3.	PC-3 xenograft model [128].
5.2 Glucoraphanin → Thiocyanates + Isothiocyanates + Indoles
5.2.1 Sulforaphane (SFN)	Isothiocyanates (ITCs)	Cruciferous vegetables	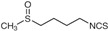	Induction of apoptosis; Induction of G2/M cell cycle arrest; Inhibition of FA metabolism.	ERK1/2 [129]; Hsp90 [130]; AR [131]; ROS [132]; FA [133]; HDACs [134].	LNCaP, DU 145 and PC-3.	TRAMP mice [134].
5.2.2 Phenethyl-Isothiocyan (PEITC)	Isothiocyanates (ITCs)	Cruciferous vegetables	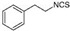	Induction of G2/M cell cycle arrest; Induction of apoptosis; Inhibition of angiogenesis.	α- and β-tubulin [135]; Bax [136]; Akt [137]; PECAM1-CD31 [138].	C4-2B, DU 145, PC-3, and LNCaP.	LNCaP xenograft model [138]; TRAMP mice [139].
5.2.3 Indole-3-Carbinol	Indoles	Cruciferous vegetables	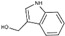	Induction of apoptosis and cell cycle arrest; Modulation of epigenetic alterations of cancer stem cells.	Bax [140]; NF-ĸB, Nrf2 [140].	LNCaP, and PC-3.

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
