# Peer review of "Phytochemicals in Inhibition of Prostate Cancer: Evidence from Molecular Mechanisms Studies"

_biomolecules, 2022, doi:10.3390/biom12091306_

Round 1

Reviewer 1 Report

The authors reviewed the key studies which indicated the beneficial effect of phytochemicals towards prostate cancer. The authors further provided specific and concise examples of relevant studies that has identified useful bioactive components with potential anti-prostate cancer capacity from various plant sources. The useful compilations uncovered in this manuscript provides inspiring insights for both the scientific community and the industry.

However, the authors should address several issues in this manuscript:

Major points:

1. The main body of the review is a little bit messy and fragmented. It would be better if the authors could correlate the stages of PCa with relevant evolution of signaling pathways/carcinogenesis processes, and correspondingly indicating the bioactivities and therapeutic potential of respective natural products.

2. The authors should consider reorganizing the category system. Currently the substances are grouped according to the plant source, however, this is confusing in some cases. For example,

a. Fruit: 2.2 Quercetin is from Apple and onion;

b. Herb: 4.8 Daidzein is from soybean which could also be considered as a vegetable;

c. Tea: There is only one component in this category, however, substances such as quercetin and gallic acid were also reported to be found in tea.

In fact, Quercetin and other compounds could be found in many plants and thus could be included in multiple categories. The authors should consider reorganizing the natural products based on the chemical structure rather than their plant source.

3. Recently, there are also some new discoveries regarding phytochemicals and marine natural products with anti-prostate cancer activity. The authors should consider citing more recent literatures.

Minor points:

1. Figure 2 is difficult to understand:

a. The authors should not only list the names of natural product near pathways, but also provide marks indicating their inhibitory or stimulatory effect.

b. The target pathway of some boxes are not clear enough, e.g., it seems that the box containing UA,TQ, FN is contacting both Akt/mTOR pathway and Ras/MEK/ERK1/2 pathway.

c. Fonts and shapes for the same component should be standard in size, e.g., AR, p53

d. Some of the labels are too small and difficult to identify, the authors should consider deleting the labels in phosphorylation and ROS symbols.

e. it will be better if the authors assign different colors for different genes/components, e.g., Cas3 & Cas9, ROS, Nrf2 & p21.

2. Table 1 is not clear and informative enough:

a. Plant source, if available, besides the common name, the authors should consider adding another column, listing the botanical name of the plant.

b. Chemical structure is not standardized, e.g., the NCS group in SFN and PEITC.

c. The authors should also adjust the sizes to make the chemical structure more standardized.

d. For Molecular mechanisms identified, the authors should at least briefly describe the effect on the listed target genes, e.g., whether the natural product is activating the enzyme or upregulating its mRNA/protein expression.

e. The authors should also indicate through which experiment models are these molecular mechanisms discovered, e.g., in vitro model, animal model, or clinical trials.

3. The English writing needs to be polished in order to make the text easier for others to understand, e.g., line 235, adequately-powered studies.

Overall, I agreed that this paper may be relevant for publication in Biomolecules. The manuscript covers the key discoveries in this field and is relatively well written, I therefore recommend reconsideration of the manuscript after major revision.

Author Response

Response: We really appreciate the reviewer's comments and suggestions. We have revised the manuscript accordingly, and the changes are highlighted in yellow throughout the manuscript.

Major points:

  1. The main body of the review is a little bit messy and fragmented. It would be better if the authors could correlate the stages of PCa with relevant evolution of signaling pathways/carcinogenesis processes, and correspondingly indicating the bioactivities and therapeutic potential of respective natural products.

Response:

Too many chemicals and corresponding inhibitory mechanisms may confuse the content. We have deleted some old and dispensable citations and re-edited the content based on the relevant evolution of carcinogenesis signaling processes. The therapeutic potential of respective natural products is indicated in the context. However, all the studies cited in this review depicted the molecular mechanism of phytochemicals against PCa cells, not involving the clinical stage of PCa.

  1. The authors should consider reorganizing the category system. Currently the substances are grouped according to the plant source, however, this is confusing in some cases. For example,
  2. Fruit: 2.2 Quercetin is from Apple and onion;
  3. Herb: 4.8 Daidzein is from soybean which could also be considered as a vegetable;
  4. Tea: There is only one component in this category, however, substances such as quercetin and gallic acid were also reported to be found in tea.

In fact, Quercetin and other compounds could be found in many plants and thus could be included in multiple categories. The authors should consider reorganizing the natural products based on the chemical structure rather than their plant source.

Response:

This is an excellent suggestion. We have elucidated the mechanism of action of the phytochemical based on their chemical structure.

  1. Recently, there are also some new discoveries regarding phytochemicals and marine natural products with anti-prostate cancer activity. The authors should consider citing more recent literatures.

Response:

We have added some latest studies about phytochemicals' anti-prostate cancer activity.

Minor points:

  1. Figure 2 is difficult to understand:
  2. The authors should not only list the names of natural product near pathways, but also provide marks indicating their inhibitory or stimulatory effect.

Response:

We have marked the inhibitory or stimulatory effect of the phytochemicals.

  1. The target pathway of some boxes are not clear enough, e.g., it seems that the box containing UA,TQ, FN is contacting both Akt/mTOR pathway and Ras/MEK/ERK1/2 pathway.

Response:

Yes, some phytochemicals inhibit PCa cell growth through the multi-pathway.

  1. Fonts and shapes for the same component should be standard in size, e.g., AR, p53

Response:

We have standardized all the same components in size, fonts, and shapes.

  1. Some of the labels are too small and difficult to identify, the authors should consider deleting the labels in phosphorylation and ROS symbols.

Response:

We have deleted the labels in phosphorylation and ROS symbols.

  1. it will be better if the authors assign different colors for different genes/components, e.g., Cas3 & Cas9, ROS, Nrf2 & p21.

Response:

We have changed accordingly.

  1. Table 1 is not clear and informative enough:
  2. Plant source, if available, besides the common name, the authors should consider adding another column, listing the botanical name of the plant.

Response:

The plant source is listed in the 3rd column of Table1

  1. Chemical structure is not standardized, e.g., the NCS group in SFN and PEITC.

Response:

We have corrected it.

  1. The authors should also adjust the sizes to make the chemical structure more standardized.

Response:

We have adjusted the size of the chemical structure to make them proportionally.

  1. For Molecular mechanisms identified, the authors should at least briefly describe the effect on the listed target genes, e.g., whether the natural product is activating the enzyme or upregulating its mRNA/protein expression.

Response:

We have added a brief description of the phytochemical's effect.

  1. The authors should also indicate through which experiment models are these molecular mechanisms discovered, e.g., in vitro model, animal model, or clinical trials.

Response:

We have added the experimental model used in these studies.

  1. The English writing needs to be polished in order to make the text easier for others to understand, e.g., line 235, adequately-powered studies.

Response: 

We have polished the entire context by the specialist.

Reviewer 2 Report

In the manuscript entitled “Natural Products in Treatment and Chemotherapy of Prostate Cancer: Evidence from Molecular Mechanisms Studies" authored by Qiongyu Hao et al., the authors present a narrative review on an important topic - prostate cancer and the therapeutical effects of natural products. Although the subject is relevant for the medical field, several major modifications are required as follows:

Point 1 -  Title - the word treatment is not the best choice since the information provided wiinth the manuscript does not refer to the actual therapeutical protocols including some products. 

Point 2 - Abstract

-the current form is rather short, and it should be developed

- several existing phrases should be modified to avoid word repetitions and ideas - eg lines 11-12 - "increasingly ...increase ....", lines 12-20 "many", 

- scientific and medical formulations are recommended - lines 12-20

"Laboratory studies" - what type of laboratory studies ? in vitro/in vivo

- "have revealed that natural compounds can affect cellular proliferation" what cell type ? tumoral cells ? 

“Many ? replace with several or a multitude of …..”natural compounds have been found to induce cell cycle arrest, promote apoptosis, inhibit cancer cell growth and suppress angiogenesis. In addition, combinatorial use of natural compounds with hormone and/or chemotherapeutic drugs seems to be a promising strategy to enhance the therapeutic effect in a less toxic manner, as suggested from pre-clinical  studies.”

- rephrase to avoid repetitions of therapeutic

- correction from with by

"In this context, we systematically reviewed the current literature of naturally occurring compounds isolated from vegetables, fruits, teas, and herbs with the relevant mechanisms of action in prostate cancer"

- replace current with currently available (several references are not current)

- consider adding the contribution to the field of knowledge.

Point 3- Introduction - it is too short and it should be developed

Also, the authors should accurately refer to prostate cancer incidence – lines 24-25

“Prostate cancer (PCa) is the most common malignancy and the second-leading cause

of cancer death worldwide for men.”

https://www.frontiersin.org/articles/10.3389/fpubh.2022.811044/full

https://exonpublications.com/index.php/exon/article/view/358/612

https://onlinelibrary.wiley.com/doi/10.1002/advs.202201859

Since the current recommended approach is early detection based on screening programs, the authors should focus on the consequences (mortality, survival rates, side effects of the medications, response to medications) – lines 44-46 can be developed.

Furthermore, the authors should consider a paragraph to better point out the novelty and the relevance of their research given previous studies

Add the method to obtain the review

Point 4 – the authors should correct - figure numbering, references style, Italic style for Latin names,  in vitro, in vivo  etc

Point 5 – the main body text is an enumeration of natural products isolated compounds with anticancer potential, their sources and associated molecular mechanisms. The authors should emphasize their particular advantages and limitations

Point 6 - Conclusion - should be modified to better underline the originality and the contribution of the study

Point 7 - the references list contains too many and several outdated studies; the authors are advised to consider only the relevant, fundamental ones.

Author Response

Response:

We really appreciate the reviewer's comments and suggestions. We have revised the manuscript accordingly, and the changes are highlighted in yellow throughout the manuscript.

Although the subject is relevant for the medical field, several major modifications are required as follows:

Point 1 -  Title

the word treatment is not the best choice since the information provided wiinth the manuscript does not refer to the actual therapeutical protocols including some products. 

Response:

We have changed the title to "Natural Products in Inhibition of Prostate Cancer: Evidence from Molecular Mechanisms Studies".

Point 2 - Abstract

-the current form is rather short, and it should be developed

- several existing phrases should be modified to avoid word repetitions and ideas - eg lines 11-12 - "increasingly ...increase ....", lines 12-20 "many", 

- scientific and medical formulations are recommended - lines 12-20

"Laboratory studies" - what type of laboratory studies ? in vitro/in vivo

- "have revealed that natural compounds can affect cellular proliferation" what cell type ? tumoral cells ? 

"Many ? replace with several or a multitude of ….." natural compounds have been found to induce cell cycle arrest, promote apoptosis, inhibit cancer cell growth and suppress angiogenesis.

In addition, combinatorial use of natural compounds with hormone and/or chemotherapeutic drugs seems to be a promising strategy to enhance the therapeutic effect in a less toxic manner, as suggested from pre-clinical  studies."

- rephrase to avoid repetitions of therapeutic

- correction from with by

"In this context, we systematically reviewed the current literature of naturally occurring compounds isolated from vegetables, fruits, teas, and herbs with the relevant mechanisms of action in prostate cancer"

- replace current with currently available (several references are not current)

- consider adding the contribution to the field of knowledge.

Response:

We have re-edited the abstract according to the above advice.

Point 3- Introduction - it is too short and it should be developed

Also, the authors should accurately refer to prostate cancer incidence – lines 24-25

"Prostate cancer (PCa) is the most common malignancy and the second-leading cause of cancer death worldwide for men."

https://www.frontiersin.org/articles/10.3389/fpubh.2022.811044/full

https://exonpublications.com/index.php/exon/article/view/358/612

https://onlinelibrary.wiley.com/doi/10.1002/advs.202201859

Since the current recommended approach is early detection based on screening programs, the authors should focus on the consequences (mortality, survival rates, side effects of the medications, response to medications) – lines 44-46 can be developed.

Furthermore, the authors should consider a paragraph to better point out the novelty and the relevance of their research given previous studies

Response:

We have re-edited the Introduction according to the above advice.

Point 4 – the authors should correct - figure numbering, references style, Italic style for Latin names,  in vitro, in vivo  etc

Response:

We have updated these accordingly.

Point 5 – the main body text is an enumeration of natural products isolated compounds with anti-cancer potential, their sources and associated molecular mechanisms. The authors should emphasize their particular advantages and limitations

Response:

We have added this information accordingly.

Point 6 - Conclusion - should be modified to better underline the originality and the contribution of the study

Response:

We have added this information accordingly.

Point 7 - the references list contains too many and several outdated studies; the authors are advised to consider only the relevant, fundamental ones.

Response:

We have re-edited the citations.

Round 2

Reviewer 1 Report

In this version, the authors addressed several issues mentioned in the previous comments, including:

Major points:

1. reorganized and re-edited the contents based on some suggestions;

2. reorganized the category system based on chemical structures;

3. added some recent literatures regarding potential phytochemicals against PCa but no potential marine natural products;

Minor points:

1. changed Figure 2 accordingly;

2. updated some parts of Table 1.

However, the authors may fail to provide the most recent version of their Table 1 in the coverletter. For Minor point 2, although they claimed to have added a brief description of the phytochemical's effect and have added the experimental model used in these studies, in the table provided, there are no such modifications.

The authors should also consider limiting the review title to “phytochemicals” rather than “natural products”.

To summarize, the authors have made several modifications to the previous version, but the manuscript fails to address the remaining issues stated above. I therefore recommend acceptance of the manuscript after minor revision.

Author Response

Response:

We appreciate the reviewer's comments and suggestions again. We have revised the manuscript accordingly, and the changes are highlighted in blue throughout the manuscript.

Reviewer #1’s comments

However, the authors may fail to provide the most recent version of their Table 1 in the coverletter. For Minor point 2, although they claimed to have added a brief description of the phytochemical's effect and have added the experimental model used in these studies, in the table provided, there are no such modifications.

Response:

Sorry for misunderstanding your comments. We have added a brief description of the phytochemical's effect and the experimental model used in these studies in table 1. The table is included in the cover letter.

The authors should also consider limiting the review title to “phytochemicals” rather than “natural products”.

Response:

This is a good suggestion. We have updated the title to “Phytochemicals in Inhibition of Prostate Cancer: Evidence from Molecular Mechanisms Studies”.

Reviewer 2 Report

point 1 - Modifications were made, but the current form has a part (lines 31-44) that maybe should be removed, not because of the setting, the content repeats several ideas.

point 2 - lines 23-28  - to produce two phrases from the last phrase (too long)  

In conclusion, the elucidation of the chemical structure-based precise mechanism of action will facilitate the identification of novel natural compounds with significant anti-cancer properties for drug development. Furthermore, the optimization of drug combination is essential to make the phytochemical as an anti-cancer agent in a non/less- toxic manner in the treatment of prostate cancer, therefore can be translated into significant health benefits for humans.

point 3 - lines 547-552 the same modifications are mentioned above

Author Response

We appreciate your comments and suggestions again. We have revised the manuscript accordingly, and the changes are highlighted in blue throughout the manuscript.

point 1 - Modifications were made, but the current form has a part (lines 31-44) that maybe should be removed, not because of the setting, the content repeats several ideas.

Response:

We have removed this part. That is a negligence, we forgot to delete. Thank you for careful reading.

point 2 - lines 23-28  - to produce two phrases from the last phrase (too long)

In conclusion, the elucidation of the chemical structure-based precise mechanism of action will facilitate the identification of novel natural compounds with significant anti-cancer properties for drug development. Furthermore, the optimization of drug combination is essential to make the phytochemical as an anti-cancer agent in a non/less- toxic manner in the treatment of prostate cancer, therefore can be translated into significant health benefits for humans.

Response:

This is good suggestion. We have re-wrote the two sentences as below:

In conclusion, the elucidation of the natural compounds’ chemical structure-based anti-cancer mechanism will facilitate the drug development, and the optimization of drug combination. Phytochemical as an anti-cancer agent in the treatment of prostate cancer can be translated into significant health benefits for humans.

point 3 - lines 547-552 the same modifications are mentioned above

Response:

This is good advice. We have simplifed these sentences as below:

However, several limitations with these phytochemicals need to be addressed on the road to a successful clinical translation. First, the low bioavailability is one of the major limitations of most phytochemicals. In vitro effective concentrations are barely achievable in vivo through oral consumption at safe doses, limiting their clinical success.